# Performance of fecal inflammatory biomarkers to identify watery shigellosis: Findings from the Enterics for Global Health (EFGH) *Shigella* surveillance study

Billy Ogwel◉[1]*, Farhana Khanam[2◉], Henry Badji[3◉], Mary Charles[4◉], Sonia Qureshi[5◉], Bri'Anna Horne[6,7], Stephanie A. Brennhofer[8], James A. Platts-Mills[8], Khandra Sears[6,7], Sharon Tennant[6,7], Sara Kim[9], Richard Omore[1], Alex O. Awuor[1], Caleb Okonji[1], Junaid Iqbal[5], Naveed Ahmed[5], Zarfishan Hussain[5], Firdausi Qadri[2], S. M. Azadul Alam Raz[2], Elias Shawon Bhuiyan[2], Pablo Penataro Yori[8], Maribel Paredes Olortegui[10], Margaret N. Kosek[8], Samba Juma Jallow[3], Bubacarr E. Ceesay[3], Bakary Conteh[3], Atusaye K. Nyirenda[4], Vitumbiko Munthali[4], Clement Lefu[4], Taufiqur Rahman Bhuiyan[2‡], Stephen Munga[1‡], M. Jahangir Hossain[3‡], Jennifer Cornick[4,11‡], Farah Naz Qamar[5‡], David Benkeser[9‡], Elizabeth T. Rogawski McQuade[9‡]*

1 Centre for Global Health Research, Kenya Medical Research Institute, Kisumu, Kenya, 2 Infectious Diseases Division, International Centre for Diarrhoeal Disease Research, Bangladesh (icddr,b), Dhaka, Bangladesh, 3 Medical Research Council Unit The Gambia at the London School of Hygiene and Tropical Medicine, Banjul, The Gambia, 4 Malawi Liverpool Wellcome Programme, Blantyre, Malawi, 5 Department of Paediatrics and Child Health, The Aga Khan University, Karachi, Pakistan, 6 Center for Vaccine Development and Global Health, University of Maryland School of Medicine, Baltimore, Maryland, United States of America, 7 Department of Medicine, University of Maryland School of Medicine, Baltimore, Maryland, United States of America, 8 Division of Infectious Diseases and International Health, University of Virginia, Charlottesville, Virginia, United States of America, 9 Department of Epidemiology, Emory University, Atlanta, Georgia, United States of America, 10 Asociación Benéfica PRISMA, Iquitos, Peru, 11 Department of Clinical Infection, Microbiology and Immunology, Institute of Infection, Veterinary and Ecological Sciences, University of Liverpool, Liverpool, United Kingdom

◉ These authors are contributed equally to this work as joint first authors.
‡ These authros are contributed equally to this work as joint senior authors.
* ogwelbill@gmail.com (OG); elizabeth.rogawski.mcquade@emory.edu (ETRM)

## Abstract

### Background

Current syndromic guidelines for diarrhea treatment miss watery *Shigella* cases, leading to undertreatment of children who may benefit. Incorporating fecal inflammatory biomarkers into diagnosis may improve case identification.

### Methods

We conducted an ancillary analysis using samples from six sites (The Gambia, Kenya, Malawi, Bangladesh, Pakistan, and Peru) from the Enterics for Global Health (EFGH)-*Shigella* surveillance study, a facility-based hybrid study of children aged 6–35 months with diarrhea. Four fecal biomarkers were quantified by enzyme-linked immunosorbent assays at enrollment: myeloperoxidase, calprotectin, neutrophil

**Data availability statement:** The EFGH datasets are publicly available through the Vivli data-sharing platform (https://search.vivli.org/doiLanding/studies/PR00011860/isLanding). The study protocol for both the primary EFGH Study and the Biomarker Sub-study were made publicly available. (https://academic.oup.com/ofid/issue/11/Supplement_1).

**Funding:** This work was funded by the Gates Foundation (grant numbers INV-044317 to ETRM; INV-016650 and INV-031791; INV-036891 to FNQ; INV-036892 to FQ; INV-028721 and INV-041730; INV-044311 to JAPM) through funding to the Enterics for Global Health (EFGH) Consortium. The funders had no role in study design, data collection and analysis, decision to publish, or preparation of the manuscript.

**Competing interests:** The authors have declared that no competing interests exist.

gelatinase-associated lipocalin (lipocalin-2), and hemoglobin. An ensemble model with leave-one-site-out cross-validation was used to predict watery shigellosis, incorporating biomarkers and nine clinical and socio-economic predictors. We compared the predictive performance of the algorithm using: a) all predictors (including biomarkers); b) all non-biomarker predictors; c) all predictors (with selected biomarkers).

## Results

Between June 2022 and August 2024, a total of 4,191/9,476 (44.2%) children presented with watery diarrhea (non-bloody) and had their whole stool tested for the biomarkers and 4,083 stool samples or rectal swabs were tested by qPCR; 735 (18.0%) had *Shigella*-attributable diarrhea by qPCR. The full model incorporating all 13 predictors achieved an area under the curve (AUC) of 0.75 [95% CI: 0.67–0.78], with a sensitivity of 0.67 and specificity of 0.75. Excluding biomarkers reduced model performance by 8% (AUC 0.67, 95% CI: 0.61-0.70). Adding hemoglobin alone improved the model's discriminatory ability by 7%, while further adding myeloperoxidase had marginal contribution (1%), and lipocalin-2 (0%) and calprotectin none (0%).

## Conclusion

Fecal hemoglobin substantially improved prediction scores for watery shigellosis. Consequently, implementation of point-of-care assays for hemoglobin could improve clinical diagnosis in these settings and inform appropriate antibiotic treatment.

### Author summary

Diarrhea remains a leading cause of illness in young children, with the bacteria *Shigella* being a major contributor. Current treatment guidelines recommend antibiotics only when diarrhea is visibly bloody. However, many children with *Shigella* infection have non-bloody, watery diarrhea and are therefore not treated, despite needing care. In this study, we explored whether simple markers of gut inflammation found in stool could improve the identification of *Shigella* infection. We analyzed stool samples from over 4,000 children across six countries in Africa, Asia, and South America and combined these markers with basic clinical information. We found that detecting small, invisible amounts of blood in stool (fecal hemoglobin) improved identification of non-bloody *Shigella* diarrhea by 7%. Other inflammation markers added little benefit. By combining fecal hemoglobin with child's age and stool frequency, we developed a simple score that matched performance of the full model enhancing its practical use. This approach outperformed current guidelines, which only treat bloody diarrhea, and was more accurate than treating most children without testing. Importantly, fecal hemoglobin tests are inexpensive and commercially available. Using them at the point of care could help health workers better target antibiotics, improve child outcomes, and promote responsible antibiotic use in low-resource settings.

## Introduction

Diarrhea is the third leading cause of death among children under five years, with the highest burden in South Asia and sub-Saharan Africa [1]. *Shigella,* diarrheagenic *Escherichia coli, Campylobacter jejuni,* and *Salmonella* are common enteroinvasive bacterial etiologies of acute diarrhea, often causing severe episodes [2]. In 2016, *Shigella* was estimated to account for 63,713 deaths among children under five years worldwide, respectively [3]. Appropriate antibiotic therapy can significantly reduce the duration and severity of bacterial diarrhea, preventing complications, hospitalizations, and secondary transmission [4]. Specifically, azithromycin has been shown to benefit children with *Shigella* diarrhea by reducing diarrheal duration and lowering the number of hospitalizations/deaths [5]. Furthermore, antibiotic treatment of *Shigella*-related moderate-to-severe diarrhea (MSD) improved short-term growth in the Global Enteric Multicenter Study (GEMS) [6] and reduced risk of persistent diarrhea in the Vaccine Impact on Diarrhea in Africa (VIDA) study [7]. However, antibiotic efficacy is dependent on accurate etiological diagnosis and local susceptibility patterns to avoid unnecessary antimicrobial exposure and resistance selection.

Clinical signs of bacterial diarrhea are often non-specific, making it difficult to distinguish watery bacterial cases from viral ones [8]. In this context, the primary barrier to appropriately targeted therapy is the lack of timely and accessible diagnostics in resource-limited settings. The gold standard, stool microbiology, has low sensitivity, particularly for *Shigella* [9,10], and a prolonged turnaround time of 48 hrs up to five days, reducing its effectiveness for acute clinical decision-making. Additionally, the use of more sensitive molecular diagnostics in the diverse range of settings where diarrhea is evaluated and managed is not feasible. In such settings, clinicians rely on current World Health Organization (WHO) syndromic-based management guidelines, which recommend the use of antibiotics in cases of clinical dysentery or suspected cholera only [11].

Antibiotic use for watery diarrhea (non-bloody) is restricted because, without diagnostic testing, dysentery is more likely to be caused by *Shigella*, making empirical treatment for watery diarrhea less beneficial, while limiting use preserves antibiotic effectiveness and supports antimicrobial stewardship. However, these restrictive guidelines lack pathogen specificity and do not recommend treatment for most bacterial diarrhea episodes, including shigellosis, which presents as watery diarrhea (non-bloody) in 40%–89% of cases [9,10,12]. Furthermore, the guidelines may not reflect local resistance patterns, often resulting in empirical over- or under-treatment, suboptimal outcomes, and antimicrobial resistance propagation [13,14]. Therefore, it is imperative to develop improved approaches at the point-of-care to identify children with bacterial diarrhea, specifically the subset with watery diarrhea who are missed by current guidelines and who will benefit from targeted antibiotic treatment.

While point-of-care (POC) tests based on polymerase chain reaction (PCR) are accurate, their cost and technical requirements limit widespread implementation in low-resource settings. Lateral flow assays offer a more feasible point-of-care alternative, although their sensitivity and specificity remain suboptimal [15]. A promising strategy to bridge this diagnostic gap is the evaluation of host inflammatory biomarkers, which could be adapted to lateral flow assays. Unlike viral pathogens, invasive pathogens like *Shigella* trigger a robust and distinctive intestinal inflammatory response, releasing proteins such as myeloperoxidase, lactoferrin, and C-reactive protein into the stool and blood [2,16]. Furthermore, *Shigella* pathogenesis involves invasion of the intestinal epithelium, triggering inflammation and mucosal damage through virulence factors [8,17]. A systematic review of inflammatory fecal biomarkers found that fecal leukocytes, red blood cells, lactoferrin, calprotectin, and myeloperoxidase were the most commonly evaluated to identify bacterial diarrhea [8]. These markers showed high sensitivity for *Shigella* and moderate sensitivity for any bacterial diarrhea, with comparable performance across biomarkers, though specificity varied by outcome. Prior studies have been limited by small sample sizes, reliance on microscopy for blood cell detection, and use of insensitive gold standards such as conventional culture [8]. More sensitive diagnostics, including Enzyme-Linked Immunosorbent Assays (ELISAs) and molecular methods, are needed to validate these biomarkers and support the development of POC tests [8]. Integrating validated biomarkers into clinical prediction algorithms, alongside clinical and epidemiological data, could provide objective, rapid indicators for watery shigellosis and bacterial diarrhea.

Although antibiotics are clearly recommended for *Shigella* and Cholera, guidance for other bacterial pathogens is less definitive. This supports the need to assess the diagnostic performance of fecal inflammatory biomarkers for detecting watery shigellosis (non-bloody) and other bacterial diarrhea. To evaluate the diagnostic performance of fecal inflammatory biomarkers to identify watery shigellosis and other bacterial diarrhea, we leveraged data from six low-resource sites of the Enterics for Global Health (EFGH)- *Shigella* surveillance study [8]. Using archived stool samples from enrolled children with medically-attended diarrhea, we measured hemoglobin, myeloperoxidase, calprotectin, and lipocalin-2 concentrations at presentation to care. We then constructed clinical prediction algorithms for diarrheal etiology, incorporating these inflammatory biomarkers and characterized their performance to identify watery shigellosis (non-bloody) and other bacterial etiologies of diarrhea.

## Methods

### Ethics statement

Prior to study initiation, site-specific study materials were approved by the Institutional Review Boards (IRBs) at each EFGH site and affiliated coordinating body, as detailed elsewhere. For the biomarker sub-study specifically, it was approved by the Emory IRB (Dec 1, 2022). Although it was considered human subjects research, it was deemed exempt from further IRB review and approval. This project meets the criteria for exemption under 45 CFR 46.104(d) [4]. Written informed consent was obtained from the caregivers of all children prior to any research procedures.

### Study setting, design and procedures

We conducted an ancillary analysis using samples from six sites (Bangladesh, Kenya, Malawi, Pakistan, Peru and The Gambia) from the Enterics for Global Health (EFGH)-*Shigella* surveillance study, a facility-based hybrid study of children aged 6–35 months with medically-attended diarrhea enrolled between June 2022- August 2024. Children aged 6–35 months presenting at the selected EFGH health facilities during working hours with a new episode of acute diarrhea (≥3 abnormally loose or watery stools in 24 hours, onset within 7 days, following at least two diarrhea-free days) with or without blood were screened for enrolment. Eligible children presented to the facility within four hours of enrollment and were not enrolled in other concurrent studies. Their caregivers provided written informed consent for participation, sampling, follow-ups and confirmed they would remain in the study catchment area for at least four months after enrolment [18]. Participants were enrolled as a consecutive series, whereby all eligible children presenting during the study period were included, minimizing selection bias and ensuring representativeness of the target population.

Trained EFGH study personnel collected information on socio-demographics, diarrhea history, recent hospitalization or healthcare-seeking, treatment received, and time of presentation to the EFGH site. A detailed clinical examination was performed, including taking vital signs, anthropometry, and assessment of dehydration and malnutrition. Supportive management followed WHO/local guidelines.

At enrollment, three rectal swabs were collected from each enrolled child for detection of enteropathogens by qPCR and *Shigella* isolation using modified Buffered Glycerol Saline and Cary-Blair transport media. Whole stool samples were collected preferably before facility discharge, and if not, caregivers were trained to collect the first stool sample at home in a stool collection kit within 24 hours of enrolment [18]. Samples were either retrieved by study staff or delivered by the caregivers (within 18 hours of collection) to the EFGH facility. Home visits by study staff were conducted only during routine working hours, and participants enrolled before a non-working day were excluded from home stool collection. Stool samples were transported under a maintained cold chain (2°C–8°C) to either the central research laboratory or to the EFGH recruitment facility of each site for further transport alongside other collected recruitment samples. Samples were aliquoted (into up to five 2-mL cryotubes (~1 g per cryotube) and stored at -80°C until processing for culture, qPCR, and biomarker analysis.

## Measurement of inflammatory biomarkers

Fecal protein inflammatory biomarkers were measured in whole stool samples using commercially available Enzyme-Linked Immunosorbent Assay (ELISA) kits, performed according to manufacturers' guidelines. Fecal calprotectin, hemoglobin, and myeloperoxidase were measured using the IDK® Calprotection ELISA [19], IDK® Hemoglobin/Haptoglobin Complex ELISA [20] and IDK® Myeloperoxidase ELISA [21] (Immundiagnostik, Bensheim, Germany), respectively. Fecal neutrophil gelatinase-associated lipocalin (lipocalin-2) was measured using the NGAL stool ELISA kit (Eagle Biosciences, New Hampshire, USA) [22]. Standards and controls were run in duplicate on each plate alongside participant samples, and biomarker concentrations were calculated from the raw optical densities using a 4-parameter curve fit to the standards using a web-accessible custom-built R-based Shiny application. Fecal biomarker measurements were performed blinded to clinical information and reference standard (TaqMan Array Cards (TAC)) results. Indeterminate results were handled using predefined repeat-testing and imputation rules, with biomarker values outside the assay range either repeated or imputed based on standardized criteria.

## Specimen testing methods - Culture and qPCR for pathogens

Enteric pathogens were detected in rectal swabs using microbiological culture to identify *Shigella* spp. [18,23] and by quantitative PCR (qPCR) to identify a panel of enteric pathogens using TAC, as previously described [24]. If rectal swabs were unavailable, qPCR was conducted on the whole stool sample. Reference standard (TAC) results were subject to strict quality control, including multiple internal and external controls and an analytical cutoff of Cq = 35. Only results with valid control performance and no critical QC flags were retained; observations with invalid or indeterminate TAC results were excluded from the analysis [24]. Reference standard, TAC assays, were performed without knowledge of clinical data or biomarker results, ensuring that pathogen detection was independent of participant characteristics or biomarker measurements.

## Statistical methods

**Outcome variables.** We included all enrolled children with watery (non-bloody) medically-attended diarrhea (MAD). Children who experienced blood in stool during their index diarrhea episode based on caregiver report or clinician assessment were excluded. We also excluded observations with missing TAC results for *Shigella* and other bacterial etiologies, as well as those with missing biomarker concentrations, to ensure that both the reference standard (TAC) and the target tests (biomarkers) were available for direct comparison within the same individuals. We had three outcome variables: i.) *Shigella* diarrhea, defined as a MAD episode attributable to *Shigella* based on a qPCR cycle threshold ($C_T$) <29.5 detected in rectal swab samples or <29.8 detected in whole stool samples; ii.) *Shigella* diarrhea identified by standard culture methods; iii.) any bacterial diarrhea, defined an MAD episode attributable to the following pathogens using the specified $C_T$ values: tEPEC ($C_T$ < 18.6), ST-ETEC ($C_T$ < 25.6), *Campylobacter jejuni/coli* ($C_T$ < 19.6), *Salmonella enterica* ($C_T$ < 31.4), or *Vibrio cholerae* ($C_T$ < 32.7) detected in rectal swab samples or whole stool samples. The methods for the attribution of specific etiologies of diarrhea in EFGH, leveraging previous studies that compared pathogen quantities between diarrhea cases and non-diarrheal controls, has been described in detail elsewhere [24].

**Predictors.** The algorithms were developed based on a total of 13 predictors. The primary inflammatory biomarkers of interest were myeloperoxidase, calprotectin, lipocalin-2, and hemoglobin concentrations transformed to a log10 scale, which were tested on whole stool specimens as detailed previously [8]. Additional clinical and socio-economic predictors, collected as part of the study procedures at enrollment [18,25] included: child age (months), height-for-age Z-score, pre-enrollment diarrhea duration, number of loose stools in the previous 24 hours, vomiting, WHO dehydration status that classifies patients into no dehydration, some dehydration, or severe dehydration, fever, site-specific indicator of the *Shigella* transmission season (defined as 3 months with the highest *Shigella* prevalence in each site), and household wealth index.

**Descriptive analysis.** We summarized the characteristics of the study population with watery diarrhea, both overall and stratified by site, using the predictors listed above. Categorical variables were reported as frequencies and percentages, while continuous variables were described using medians and interquartile ranges (IQR). We also assessed the diagnostic performance of each of the 4 biomarkers individually using ROC analysis [26] to assess their ability to predict the three outcomes. For each biomarker-outcome combination, we applied the cutpointr package to calculate the optimal cutoff point based on maximization of the Youden's index, which weighs sensitivity and specificity equally. Continuous biomarker log values were used in model development but the dichotomized version based on the optimal cutoff point was used in developing the clinical score.

**Model development.** For two of the outcomes (*Shigella* and any bacterial diarrhea by qPCR), we developed prediction algorithms under 3 scenarios: i.) using all 13 predictors, ii.) using all clinical predictors excluding the biomarkers, and iii.) starting with a model without biomarkers then incrementally adding each biomarker in order based on the variable importance derived from the full model including all the 13 predictors. The first scenario aimed to evaluate the overall predictive performance of the biomarkers combined with socio-demographic and clinical characteristics in predicting cases of watery diarrhea attributed to *Shigella* and any bacterial infection, respectively. The second and third scenarios were designed to assess the individual and collective contribution of the biomarkers to the predictive performance of each model.

To develop the models, we implemented a SuperLearner ensemble approach utilizing diverse algorithms as base learners to enhance the model's adaptability to non-linear patterns in the data, including generalized linear models (SL. glm), random forests (SL.randomForest), support vector machines (SL.ksvm), gradient boosting (SL.xgboost), multivariate adaptive regression splines (SL.earth), and neural networks (SL.nnet) [27,28]. The SuperLearner ensembling approach is a machine learning technique built on the theory of cross-validation that aggregates multiple base learners by finding the optimal combination of algorithms that minimize the cross-validated risk to improve accuracy and robustness [28]. Furthermore, during the model development and validation we executed a leave-one-site-out approach iterating over each study site, whereby data from all sites except the target site was used for model training. To address potential class imbalance in the outcome variables, we applied up-sampling to the minority class in the training data using the caret::upSample function [29]. For each base learner within the SuperLearner ensemble, custom wrapper functions were implemented to perform up-sampling prior to model fitting. The resulting up-sampled datasets were then used to train the respective learners, setting cross-validation with 10 folds within the training data to reduce overfitting and ensure generalizability. Predictions were generated for the validation data (left-out site) to assess model performance, allowing us to evaluate each model's external validity across different geographic locations. We used a model agnostic variable importance measure computed via permutation using the DALEX package to provide insights into predictors contributing most to model performance [30].

We applied bootstrapping with 1,000 resamples to the site-specific validation data to calculate percentile-based confidence intervals for the performance metrics [31], including sensitivity, specificity, positive predictive value (PPV), negative predictive value (NPV), and area under the ROC curve (AUC). For each bootstrap iteration: i.) cut points were determined first by maximizing Youden's Index and second by maximizing sensitivity with specificity constrained to at least 80%, ii.) performance metrics were calculated for each cut point, and iii.) the 2.5th and 97.5th percentiles of the bootstrap estimates were used to construct 95% confidence intervals (CIs) for each metric. Overall estimates of the performance metrics were calculated by averaging each bootstrap iteration of the validation site-specific estimates across sites and similarly constructing percentile-based confidence intervals. For the incremental model comparison (scenario iii), site-specific and overall mean difference and 95% confidence intervals (CI) were calculated for AUC, sensitivity, and specificity between successive models based on 1,000 bootstrap resamples on the validation data.

Given the documented differences in the epidemiology of *Shigella* between infants and older children [32], we performed a sensitivity analysis to predict *Shigella* by both qPCR and culture outcomes for these age groups separately: infants (6–11 months) and older children (12–35 months). We additionally performed a sensitivity analysis stratified by

symptom duration at enrollment (≤2 days vs ≥3 days) to evaluate whether disease progression from early watery diarrhea to later inflammatory stages influenced biomarker performance. These sensitivity analyses used the same modeling approach, replacing the leave-one-site-out cross-validation strategy with an 80%:20% data split for training and testing given the reduced sample size in each stratum. This alternative split sampling approach was also applied to the *Shigella* diarrhea by standard culture outcome due to the limited number of events.

To leverage the model outputs for quick and practical clinical decision-making, we used the AutoScore framework [33] to develop a parsimonious clinical score based on age, dichotomized hemoglobin variable based on optimal cut-off and stool frequency—the two most important predictors and a marker of severity. This was achieved through an iterative process that prioritized parsimony and performance. We also explored the performance of hemoglobin alone in predicting watery shigellosis. The reporting of results followed the Standards for Reporting Diagnostic accuracy studies (STARD) guidelines.

Descriptive analysis and predictive modeling were performed using R version 4.4.1 (R Foundation for Statistical Computing, Vienna, Austria).

## Results

### Patient characteristics

During the study period, 4,191/4,903 (85.5%) of enrolled children who had a whole stool sample collected, presented with watery diarrhea (non-bloody). Among the children with watery diarrhea, 108 (2.6%) did not have qPCR results and were excluded from further analysis.

Of the 4,083 children included in the analysis, more than half were male (n = 2,278, 55.8%), with a median age of 13 months (interquartile range [IQR]: 9–19 months). Additionally, 38 (0.9%) of the children had severe dehydration, while 902 (22.1%) had some dehydration. The prevalence of *Shigella* attributable diarrhea by qPCR and by culture were 18.0% (n = 735) and 7.3% (n = 299), respectively. Moreover, the prevalence of MAD attributable to non-*Shigella* bacterial etiology was 16.5% (n = 675) (Table 1). Site-level heterogeneity was observed across all variables (p < 0.001), except for sex (p = 0.741).

### Individual diagnostic performance of biomarkers

Hemoglobin demonstrated the highest discriminatory ability in predicting *Shigella* attributable diarrhea by qPCR (AUC = 0.65 [95% CI: 0.63-0.67]), while lipocalin-2 showed the lowest performance (AUC = 0.59 [95% CI: 0.55-0.61]) (Fig 1). Similarly, for *Shigella* detection by culture, hemoglobin had the highest discriminatory ability (AUC = 0.70 [95%CI: 0.67-0.74]), while lipocalin-2 had the lowest performance (AUC = 0.59 [95%CI: 0.56-0.61]) (*Fig A in* S1 Appendix). For non-*Shigella* bacterial diarrhea, biomarkers showed limited predictive value, with AUCs approximating random chance (0.51–0.54) (*Fig B in* S1 Appendix).

### Model performance in the prediction of watery shigellosis

When predicting *Shigella* attributable diarrhea by qPCR, the model incorporating all 13 predictors achieved an AUC of 0.75 [95% CI: 0.67–0.78], with a sensitivity of 0.67 and specificity of 0.75, using the cutoff determined by Youden's Index. Additionally, the PPV and NPV were 0.35 and 0.92 respectively. Model performance varied across sites, with the highest AUC observed in Kenya (0.78) and the lowest in Peru (0.66). Excluding the biomarkers resulted in a 8% reduction in model performance, with the AUC dropping to 0.67 [95% CI: 0.61-0.70]. Sensitivity improved to 0.70, while specificity dropped to 0.61 (Table 2). The Receiver operating characteristic (ROC) curves for predicting *Shigella*-attributable diarrhea are consistently higher and steeper across sites when biomarkers are included, with the strongest gains seen in Kenya and Malawi Fig 2. Variable importance had age and hemoglobin emerging as the strongest predictors, followed by season, myeloperoxidase, calprotectin, and lipocalin-2 in the top six (Fig 3).

**Table 1. Characteristics of children aged 6-35 months presenting with watery diarrhea who had a whole stool tested for inflammatory biomarkers in 6 EFGH country sites, 2022-2024.**

| Variable | Category | Bangladesh | Kenya | Malawi | Pakistan | Peru | The Gambia | Overall | P-value* |
|---|---|---|---|---|---|---|---|---|---|
| | | n = 1,037 | n = 749 | n = 586 | n = 485 | n = 383 | n = 843 | N = 4083 | |
| | | n (%) | n (%) | n (%) | n (%) | n (%) | n (%) | n (%) | |
| Age months | Median [IQR] | 13 [9-19] | 12 [8-19] | 14 [9-20] | 15 [10-22] | 15 [11-21] | 14 [10-20] | 13 [9-20] | <0.001 |
| Age category | 6-11m | 454 (43.8) | 337 (45.0) | 245 (41.8) | 173 (35.7) | 113 (29.5) | 303 (35.9) | 1625 (39.8) | <0.001 |
| | 12-17m | 265 (25.6) | 185 (24.7) | 143 (24.4) | 118 (24.3) | 121 (31.6) | 230 (27.3) | 1062 (26.0) | |
| | 18-23m | 177 (17.1) | 108 (14.4) | 104 (17.7) | 98 (20.2) | 92 (24.0) | 189 (22.4) | 768 (18.8) | |
| | 24-35m | 141 (13.6) | 119 (15.9) | 94 (16.0) | 96 (19.8) | 57 (14.9) | 121 (14.4) | 628 (15.4) | |
| Sex | Male | 589 (56.8) | 416 (55.5) | 316 (53.9) | 271 (55.9) | 224 (58.5) | 462 (54.8) | 2278 (55.8) | 0.741 |
| | Female | 448 (43.2) | 333 (44.5) | 270 (46.1) | 214 (44.1) | 159 (41.5) | 381 (45.2) | 1805 (44.2) | |
| Wealth Quintiles | Quintile 1- Least wealthy | 0 (0) | 15 (2.0) | 61 (10.4) | 177 (36.5) | 162 (42.3) | 204 (24.2) | 619 (15.2) | <0.001 |
| | Quintile 2 | 65 (6.3) | 125 (16.7) | 188 (32.1) | 175 (36.1) | 147 (38.4) | 352 (41.8) | 1052 (25.8) | |
| | Quintile 3 | 206 (19.9) | 203 (27.1) | 240 (41.0) | 62 (12.8) | 72 (18.8) | 228 (27.0) | 1011 (24.8) | |
| | Quintile 4 | 338 (32.6) | 305 (40.7) | 76 (13.0) | 55 (11.3) | 2 (0.5) | 59 (7.0) | 835 (20.5) | |
| | Quintile 5- Wealthiest | 428 (41.3) | 101 (13.5) | 21 (3.6) | 16 (3.3) | 0 (0) | 0 (0) | 566 (13.9) | |
| Maximum number of loose stools[β] | Median [IQR] | 8 [6-12] | 4 [3-6] | 6 [5-8] | 6 [5-8] | 4 [3-6] | 5 [4-7] | 6 [4-8] | <0.001 |
| Diarrhea days[β] | Median [IQR] | 4 [3-5] | 2 [2-3] | 2 [2-3] | 2 [1-3] | 3 [1-4] | 2 [2-3] | 3 [2-4] | <0.001 |
| Vomiting[β] | Yes | 388 (37.4) | 338 (45.1) | 266 (45.4) | 141 (29.1) | 184 (48.0) | 342 (40.6) | 1659 (40.6) | <0.001 |
| Dehydration[β] | Severe | 2 (0.2) | 19 (2.5) | 1 (0.2) | 1 (0.2) | 2 (0.5) | 13 (1.5) | 38 (0.9) | <0.001 |
| | Some | 122 (11.8) | 338 (45.1) | 23 (3.9) | 85 (17.5) | 289 (75.5) | 45 (5.3) | 902 (22.1) | |
| | None | 913 (88.0) | 392 (52.3) | 562 (95.9) | 399 (82.3) | 92 (24.0) | 785 (93.1) | 3143 (77) | |
| Fever[β] | Yes | 103 (9.9) | 266 (35.5) | 45 (7.7) | 53 (10.9) | 52 (13.6) | 294 (34.9) | 813 (19.9) | <0.001 |
| Height-age-Z score | Median [IQR] | -1.2 [-1.9--0.4] | -0.9 [-1.8--0.3] | -1.2 [-2--0.5] | -1.6 [-2.5--0.8] | -1.1 [-1.8--0.4] | -1.2 [-1.9--0.4] | -1.2 [-2--0.4] | <0.001 |
| Calprotectin (ng/mL) | Median [IQR] | 4.7 [4.2-5.2] | 5.1 [4.6-5.4] | 4.9 [4.5-5.3] | 5.1 [4.7-5.4] | 5.4 [5.1-5.8] | 5.1 [4.7-5.5] | 5 [4.5-5.4] | <0.001 |
| Hemoglobin (µg/g) | Median [IQR] | 0 [0-0.4] | 0.4 [0-2.5] | 0 [0-0.6] | 0 [0-1] | 0.1 [0-0.8] | 0.4 [0-4.3] | 0.1 [0-1.3] | <0.001 |
| Myeloperoxidase (ng/mL) | Median [IQR] | 703.8 [391.4-1640.1] | 1554.4 [733.8-3054.1] | 1442.3 [722-3010.3] | 1265.1 [532.2-3106.7] | 2257.6 [1009.4-4816.3] | 1640.5 [745.7-3884.3] | 1302.6 [571-3053.8] | <0.001 |
| Lipocalin-2 (µg/g) | Median [IQR] | 23.6 [8.8-48.2] | 29.6 [16.3-54.8] | 19 [6-40.6] | 3.9 [1.6-7.5] | 34.2 [16.5-53.4] | 42.3 [22.7-71.5] | 24.7 [8.2-50] | <0.001 |
| *Shigella* attributed by qPCR | Yes | 227 (21.9) | 93 (12.4) | 50 (8.5) | 107 (22.1) | 63 (16.4) | 195 (23.1) | 735 (18.0) | <0.001 |
| *Shigella* detected by culture | Yes | 106 (10.2) | 42 (5.6) | 35 (6.0) | 39 (8.0) | 21 (5.5) | 56 (6.6) | 299 (7.3) | <0.001 |
| Any Bacteria attributed by qPCR | Yes | 232 (22.4) | 79 (10.5) | 57 (9.7) | 96 (19.8) | 13 (3.4) | 198 (23.5) | 675 (16.5) | <0.001 |

* P-values for categorical variables were estimated using Chi-square or Fisher's exact test (with Monte Carlo simulation for sparse data). Continuous variables were compared using the Kruskal-Wallis test.

[β]Severity variables represent symptoms up to enrolment.

The model performance when the cutoff was determined by constraining specificity to 0.80 are shown in *Table A in S1 Appendix*, with sensitivity lower at 0.35-0.64. For *Shigella* detection by culture, the model achieved an AUC of 0.79 (95% CI: 0.74-0.85), a sensitivity of 0.69, and a specificity of 0.58 using the cutoff determined by Youden's Index (*Table B in S1 Appendix*)—performance comparable to the prediction of *Shigella*-attributable diarrhea by qPCR. When the *Shigella* transmission season indicator was excluded, the overall models discriminatory dropped by 3% (AUC = 0.72, 95%CI: 0.60-0.78) (*Table C in S1 Appendix*). When stratified by age, model performance for predicting qPCR-confirmed *Shigella*-attributable diarrhea was broadly consistent, though lowest among infants. The AUC was ~6% higher in older children, increasing

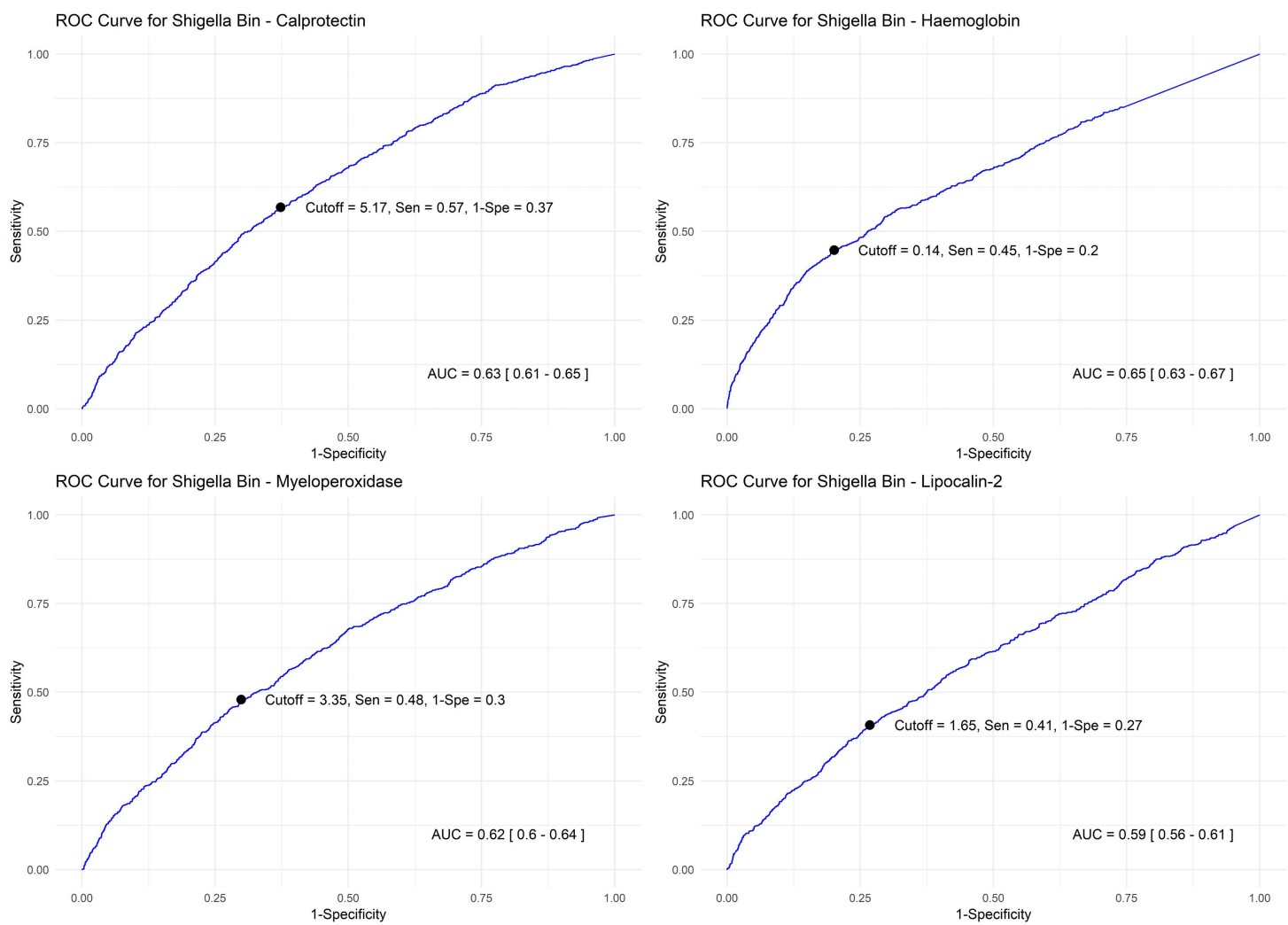

**Fig 1. Diagnostic characteristics of the biomarkers individually to identify watery diarrhea attributed to *Shigella* by qPCR.**

from 0.72 (95% CI: 0.65–0.77) in those aged 6–11 months to 0.78 in both the 12–23 months (95% CI: 0.74–0.82) and 24–35 months (95% CI: 0.71–0.85) age groups (*Table D in S1 Appendix*). When stratified by duration of symptom onset the model performance was comparable: ≤ 2 days (AUC = 0.79, 95% CI: 0.73-0.84) and ≥3 days (AUC = 0.77, 95% CI: 0.73-0.81) (*Table E in S1 Appendix*).

In the incremental models, hemoglobin improved the model's discriminatory ability by 7% and this mean difference in AUC was statistically significant. Moreover, hemoglobin significantly improved performance in all sites except Peru. However, the subsequent addition of myeloperoxidase, lipocalin-2 and calprotectin yielded marginal contributions at 1%, 0% and 0%, respectively (Table 3). The mean differences in model sensitivities and specificities are shown in *Table E in S1 Appendix*.

## Model performance in the prediction of watery bacterial diarrhea

The prediction of watery bacterial diarrhea (AUC = 0.55, 95% CI: 0.51-0.59) showed minimal predictive performance with the incorporation of all 13 predictors resulting in only a 3% AUC improvement compared to the model excluding

**Table 2. Model Performance in the prediction of watery diarrhea attributed to *Shigella* using Youden's index cutoff.**

| Validation site | Sensitivity [95% CI] | Specificity [95% CI] | PPV [95% CI] | NPV [95% CI] | AUC [95% CI] |
|---|---|---|---|---|---|
| **Including biomarkers and clinical and socio-demographic predictors** | | | | | |
| Bangladesh | 0.58 [0.52-0.71] | 0.78 [0.65-0.82] | 0.43 [0.35-0.47] | 0.87 [0.85-0.89] | 0.72 [0.69-0.74] |
| Kenya | 0.72 [0.56-0.80] | 0.72 [0.67-0.89] | 0.28 [0.23-0.41] | 0.95 [0.93-0.96] | 0.78 [0.73-0.82] |
| Malawi | 0.62 [0.52-0.81] | 0.87 [0.67-0.90] | 0.29 [0.16-0.38] | 0.96 [0.95-0.98] | 0.77 [0.71-0.83] |
| Pakistan | 0.64 [0.52-0.78] | 0.77 [0.61-0.85] | 0.43 [0.34-0.52] | 0.88 [0.85-0.91] | 0.74 [0.69-0.78] |
| Peru | 0.85 [0.63-0.93] | 0.46 [0.38-0.70] | 0.24 [0.19-0.30] | 0.94 [0.90-0.97] | 0.66 [0.61-0.72] |
| The Gambia | 0.70 [0.61-0.84] | 0.71 [0.57-0.79] | 0.42 [0.35-0.49] | 0.89 [0.86-0.93] | 0.76 [0.73-0.79] |
| Overall | 0.67 [0.59-0.83] | 0.75 [0.49-0.84] | 0.35 [0.25-0.43] | 0.92 [0.87-0.96] | 0.75 [0.67-0.78] |
| **Excluding biomarkers** | | | | | |
| Validation site | Sensitivity [95% CI] | Specificity [95% CI] | PPV [95% CI] | NPV [95% CI] | AUC [95% CI] |
| Bangladesh | 0.67 [0.51-0.76] | 0.64 [0.55-0.78] | 0.34 [0.30-0.41] | 0.87 [0.84-0.90] | 0.68 [0.65-0.71] |
| Kenya | 0.81 [0.60-0.94] | 0.50 [0.36-0.70] | 0.19 [0.16-0.24] | 0.95 [0.92-0.98] | 0.68 [0.63-0.72] |
| Malawi | 0.76 [0.45-0.88] | 0.57 [0.44-0.84] | 0.14 [0.11-0.22] | 0.96 [0.94-0.98] | 0.67 [0.62-0.73] |
| Pakistan | 0.53 [0.35-0.82] | 0.70 [0.37-0.85] | 0.33 [0.27-0.41] | 0.84 [0.81-0.88] | 0.63 [0.59-0.68] |
| Peru | 0.91 [0.76-0.98] | 0.35 [0.26-0.50] | 0.22 [0.18-0.25] | 0.95 [0.91-0.99] | 0.61 [0.56-0.66] |
| The Gambia | 0.62 [0.52-0.84] | 0.71 [0.46-0.80] | 0.38 [0.31-0.45] | 0.86 [0.84-0.91] | 0.71 [0.67-0.74] |
| Overall | 0.70 [0.55-0.89] | 0.61 [0.38-0.69] | 0.28 [0.15-0.38] | 0.91 [0.84-0.96] | 0.67 [0.61-0.70] |

PPV- Positive Predictive value; NPV-Negative Predictive value; AUC- Area under the Curve.

biomarkers (AUC = 0.52, 95% CI: 0.51-0.56). (*Table G in S1 Appendix*). Additionally, using Youden's index, the model achieved a sensitivity of 0.55, specificity of 0.51, PPV of 0.16, and NPV of 0.88. The variable importance ranking for watery bacterial diarrhea had season and stool frequency emerging as the strongest predictors, followed by age, myeloperoxidase, hemoglobin, and baseline HAZ in the top six (*Fig C in S1 Appendix*). The ROC curves for predicting watery bacterial diarrhea are presented in *Fig D in S1 Appendix*, comparing a model with all predictors (Scenario I) to a model excluding biomarkers (Scenario II). In the incremental models, only lipocalin-2 (2%) and calprotectin (1%) improved the model's discriminatory ability albeit minimal. Hemoglobin and myeloperoxidase did not improve model performance (0%) (*Table H in S1 Appendix*). None of these improvements were statistically significant. The mean differences in model sensitivities and specificities are shown in *Table I in S1 Appendix*.

## Clinical score development

To facilitate clinical application, we developed a simple 10-point clinical score based on age, dichotomized hemoglobin, and stool frequency—the two most important predictors and a marker of severity. This score exhibited promising performance (AUC = 0.75, 95% CI: 0.70-0.79; sensitivity = 0.66, specificity = 0.71) and had a treatment threshold of 6 points (Table 4). Hemoglobin alone, dichotomized at a threshold of 0.14 log10 units, showed modest discriminatory ability (AUC = 0.65, 95% CI: 0.63-0.67), with a sensitivity of 0.45 and specificity of 0.80.

## Discussion

We evaluated the performance of inflammatory biomarkers to identify watery shigellosis and other bacterial etiologies of diarrhea. Our predictive models for watery shigellosis outperformed current WHO syndromic guidelines (which recommend treatment only for dysentery) in sensitivity, while demonstrating greater specificity than the current clinical practice of empirically treating the majority of cases. Additionally, the models showed superior predictive accuracy for watery shigellosis compared to other bacterial etiologies. Model performance varied across sites with an AUC of 0.66 in Peru and

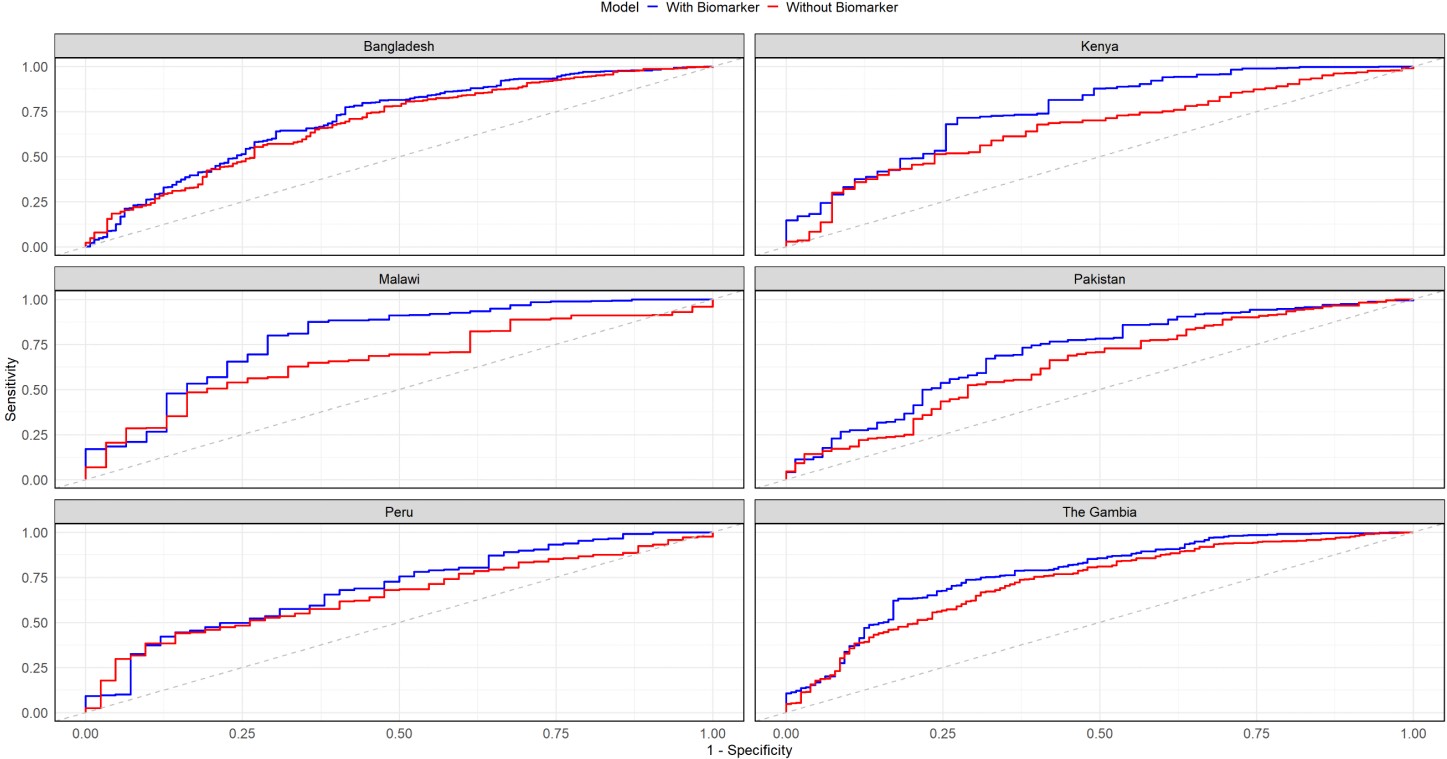

**Fig 2. ROC curves for watery shigellosis prediction models with and without fecal biomarkers stratified by site.**

0.78 in Kenya. This possibly reflects geographic differences in biomarker expression, pathogen distribution, or co-infections. Importantly, the addition of hemoglobin alone to the clinical prediction algorithm performed best, with its inclusion significantly improving the model's discriminatory ability by 7%. The practical utility of this finding is bolstered by the availability of commercial POC tests that detect fecal hemoglobin (iFOBT/FIT). To further maximize the feasibility of practical application, we developed a 10-point clinical score incorporating age, hemoglobin, and stool frequency, which showed promising performance (AUC = 0.75) when applying a treatment threshold of six points.

The above findings highlight the potential of integrating inflammatory biomarkers with clinical data to improve diagnostic accuracy for watery shigellosis in children. Notably, HAEM emerged as the most predictive biomarker, followed by MPO, CAL, and NGAL. These results suggest that blood-associated markers, such as HAEM, are reliable indicators of shigellosis, as they reflect intestinal damage caused by bacteria. This is consistent with previous studies showing that *Shigella* infections lead to blood leakage into stool [34,35]. NGAL was the least predictive biomarker in this study, which contrasts with studies that have found it to be a useful biomarker in inflammatory bowel disease and other gastrointestinal infections [8]. One plausible explanation is that NGAL's role in *Shigella* infections may be limited, or that its levels fluctuate with disease severity and host factors [36]. While these biomarkers help distinguish bacterial from viral diarrhea, their individual performance remains moderate. Earlier studies often relied on culture-based diagnostics, which had lower sensitivity compared to qPCR [35]. The current study addressed this limitation by integrating molecular diagnostics, offering a more robust and comprehensive assessment of biomarker performance. Nevertheless, differences in study designs and patient populations across previous studies may explain the variability observed in diagnostic accuracy.

Generally, an AUC of 0.5 indicates no discriminatory ability (the test cannot distinguish between those with and without the condition), 0.7–0.8 is considered acceptable, 0.8–0.9 is excellent, and above 0.9 is outstanding [37]. Our model's

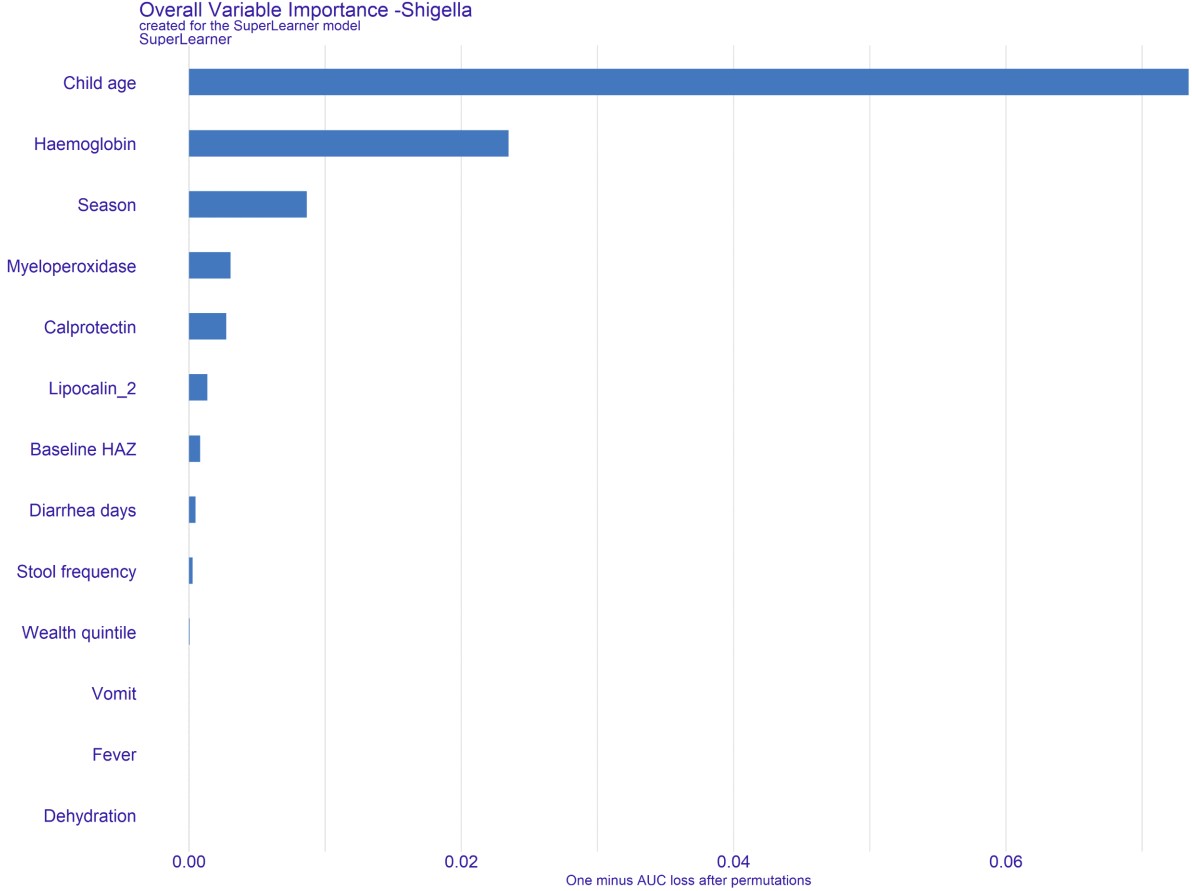

Fig 3. Variable importance in the prediction of watery diarrhea attributed to *Shigella* by qPCR.

Table 3. Comparison of bootstrapped AUCs for watery shigellosis prediction with iterative biomarker inclusion.

| Validation site | Incremental mean difference in AUC (95% CI) | | | |
|---|---|---|---|---|
| | Clinical predictors＋HAEM vs. Clinical predictors | Clinical predictors＋HAEM and MPO vs. Clinical predictors＋HAEM | Clinical predictors＋HAEM, MPO, CAL vs. Clinical predictors＋HAEM＋MPO | Clinical predictors＋HAEM, MPO, CAL, NGAL vs. Clinical predictors＋HAEM＋MPO＋CAL |
| Bangladesh | 0.05 [0.02-0.08] | 0.00 [-0.02-0.01] | 0.01 [0.00-0.03] | 0.02 [0.01-0.03] |
| Kenya | 0.08 [0.05-0.12] | 0.02 [0.00-0.04] | 0.00 [-0.02-0.01] | 0.00 [-0.01-0.01] |
| Malawi | 0.11 [0.08-0.15] | 0.00 [-0.03-0.03] | 0.00 [-0.01-0.02] | -0.02 [-0.03-0.00] |
| Pakistan | 0.09 [0.06-0.12] | 0.01 [-0.01-0.04] | -0.01 [-0.03-0.00] | 0.00 [-0.01-0.02] |
| Peru | 0.03 [-0.02-0.08] | 0.00 [-0.03-0.03] | 0.01 [-0.01-0.03] | 0.02 [0.00-0.04] |
| The Gambia | 0.05 [0.02-0.07] | 0.01 [-0.01-0.02] | 0.00 [-0.01-0.01] | 0.00 [-0.01-0.01] |
| Overall | 0.07 [0.01-0.13] | 0.01 [-0.02-0.03] | 0.00 [-0.02-0.02] | 0.00 [-0.03-0.03] |

HAEM- Hemoglobin; MPO- Myeloperoxidase; CAL- Calprotectin, NGAL- Lipocalin-2.

**Table 4. A three-variable 10-point clinical score for watery shigellosis.**

| Variable | Category | Point |
|---|---|---|
| Age category | 6-11 months | 1 |
| | 12-17 months | 2 |
| | 18-35 months | 5 |
| Hemoglobin | Low (<0.14 log10 concentration) | 0 |
| | High (≥0.14 log10 concentration) | 3 |
| Stool Frequency | 3 stools | 0 |
| | ≥4 stools | 2 |

Treatment threshold: ≥6.

**AUC**: 0.75 [0.70.4-0.79]; **Sen**: 0.66 [0.58-0.74]; **Spe**c: 0.71 [0.68-0.74]; **PPV**: 0.35 [0.31-0.39]; **NPV**: 0.90 [0.88-0.92]

performance is acceptable and comparable to prior prediction tools (AUCs:0.71-0.86) [38], but it makes distinct contributions by addressing previous key limitations. First, Brintz et al.'s model achieved an AUC of 0.825 using clinical variables and recent diagnoses with a focus on viral versus non-viral subsets, which limited its applicability and generalizability [39]. Second, Ahmed et al. utilized clinical prediction rules and achieved good predictive accuracy (AUC~0.80), but their inclusion of dysentery, a condition already targeted by syndromic-based guidelines, limits their added value for antibiotic stewardship [40]. A more recent study that prioritized antibiotic stewardship by guiding diagnostics with simple Clinical Prediction Rules showed a high predictive ability with an AUC of 0.80 [40]. Even though this model performed marginally better than our prediction model, this study included dysentery for which guidelines already recommend treatment and therefore a treatment decision tool is not needed. In contrast, our biomarker-informed approach, which incorporates clinical predictors and excludes visible cases of dysentery but not co-infections, provides a broader and more generalizable study population for predicting watery shigellosis, and thus offers a greater potential to guide treatment decisions in diverse epidemiological settings.

To enhance the model's practical utility, we developed a 10-point clinical score based on age, HAEM, and stool frequency that achieved promising performance (AUC=0.75) at a treatment threshold of ≥6 points. Beyond this clinical score, there is also potential to integrate our biomarker-based model into mobile clinical decision support systems (CDSS) for point-of-care use, as has been demonstrated in other studies [41,42].

Our study had several strengths. First, the study was conducted in multiple sites that participated in the EFGH study which enhances the generalizability and robustness of the findings. Cross-validation by site directly evaluated generalization performance. Second, to ensure consistency and reliability of results across the participating sites, training was conducted at each site by experts to standardize techniques and all analysis was conducted centrally, thereby minimizing variability and improving the reliability of the results. Third, our study used qPCR to attribute diarrhea etiology, which demonstrates greater sensitivity than culture.

Our study was limited by the fact that not all diarrhea cases had whole stool samples collected, and fewer were tested in several sites than the initial target of 80%. This reduction in sample size may have impacted the statistical power and generalizability of the findings. Additionally, to increase the number of samples, some were collected at home within 24 hours of diarrheal case enrollment. While this approach helped increase the sample size, the home collection process may have reduced the reliability of the results due to potential variations in handling and storage conditions and potential degradation of biomarkers.

Inclusion of biomarkers in our study offered a biologically grounded tool that may further refine antibiotic targeting, especially in settings with high prevalence of mixed infections. The slightly lower predictive contribution of MPO and CAL in our model, compared to previous reports [16,43] could be due to regional differences in inflammation response or assay

variability, underscoring the importance of external validation across diverse populations. These studies collectively illustrate that integrating clinical, epidemiological, and biomarker data tailored to resource availability can optimize diarrhea diagnostics and support rational evidence-based antibiotic use.

Overall, our study demonstrated the diagnostic potential of inflammatory biomarkers in identifying watery shigellosis and identifies important limitations of symptom-based models. The results support including a POC fecal hemoglobin test within triage algorithms for children with watery diarrhea. As a low-cost, commercially available assay can effectively flag likely invasive/inflammatory etiologies, to guide targeted antibiotic management. More broadly, integrating biomarkers, especially hemoglobin, with clinical predictors and mobile CDSS may represents a more comprehensive and scalable strategy for improving pediatric diarrhea diagnostics in low-resource settings.

## Supporting information

**S1 Appendix. Fig A. Diagnostic characteristics of the biomarkers individually to identify watery diarrhea attributed to *Shigella* by culture. Fig B.** Diagnostic characteristics of the biomarkers individually to identify watery bacterial diarrhea by qPCR. **Fig C.** Variable importance in the prediction of watery bacterial diarrhea. **Fig D.** ROC curves for watery bacterial diarrhea prediction models with and without fecal biomarkers. **Table A.** Model Performance in the prediction of watery diarrhea attributed to *Shigella* when constraining specificity to 0.8. **Table B.** Model performance in the prediction of *Shigella* attributable diarrhea by culture. **Table C.** Model Performance in the prediction of watery diarrhea attributed to *Shigella* when *excluding Shigella* transmission season indicator. **Table D.** Model performance in the prediction of *Shigella* attributable diarrhea by qPCR stratified by age. **Table E.** Model performance in the prediction of *Shigella* attributable diarrhea by qPCR stratified by symptom duration. **Table F.** Comparison of bootstrapped sensitivities and specificities for watery shigellosis prediction with iterative biomarker inclusion. **Table G.** Model Performance in the prediction of watery bacterial diarrhea by qPCR. **Table H.** Comparison of bootstrapped AUCs for watery bacterial diarrhea prediction with iterative biomarker inclusion. **Table I.** Comparison of bootstrapped sensitivities and specificities for watery bacterial diarrhea prediction with iterative biomarker inclusion.
(DOCX)

## Acknowledgments

The authors gratefully acknowledge the children and their families who participated in these studies, as well as the clinical, field, and laboratory teams whose commitment and hard work made this research possible. We would like to especially acknowledge Patricia Pavlinac, the overall PI for the EFGH study, and Eric Houpt, who oversaw the consortium's molecular diagnostics activities. We also thank the physicians, administrators, and health officials at each country site for providing the facilities, and support essential to the successful conduct of the study.

**Disclosure:** The findings and conclusions in this article are those of the authors and do not necessarily represent the official position of the Kenya Medical Research Institute, Emory University or partnering institutions.

## Author contributions

**Conceptualization:** Billy Ogwel, Elizabeth T Rogawski McQuade.

**Data curation:** Stephanie A Brennhofer.

**Formal analysis:** Billy Ogwel, Sara Kim, David Benkeser, Elizabeth T Rogawski McQuade.

**Funding acquisition:** Elizabeth T Rogawski McQuade.

**Investigation:** James A Platts-Mills, Sharon Tennant, Richard Omore, Firdausi Qadri, Margaret N Kosek, Stephen Munga, M. Jahangir Hossain, Jennifer Cornick, Farah Naz Qamar, Elizabeth T Rogawski McQuade.

**Methodology:** Billy Ogwel, Bri#39;Anna Horne, Stephanie A Brennhofer, James A Platts-Mills, Khandra Sears, Sharon Tennant, Alex O. Awuor, Taufiqur Rahman Bhuiyan, Stephen Munga, Jennifer Cornick, Elizabeth T Rogawski McQuade.

**Writing – original draft:** Billy Ogwel, Farhana Khanam, Henry Badji, Mary Charles, Sonia Qureshi, Elizabeth T Rogawski McQuade.

**Writing – review & editing:** Billy Ogwel, Farhana Khanam, Henry Badji, Mary Charles, Sonia Qureshi, Bri#39;Anna Horne, Stephanie A Brennhofer, James A Platts-Mills, Khandra Sears, Sharon Tennant, Sara Kim, Richard Omore, Alex O. Awuor, Caleb Okonji, Junaid Iqbal, Naveed Ahmed, Zarfishan Hussain, Firdausi Qadri, S.M. Azadul Alam Raz, Elias Shawon Bhuiyan, Pablo Penataro Yori, Maribel Paredes Olortegui, Margaret N Kosek, Samba Juma Jallow, Bubacarr E Ceesay, Bakary Conteh, Atusaye K. Nyirenda, Vitumbiko Munthali, Clement Lefu, Taufiqur Rahman Bhuiyan, Stephen Munga, M. Jahangir Hossain, Jennifer Cornick, Farah Naz Qamar, David Benkeser, Elizabeth T Rogawski McQuade.

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
