## [Decision Letter · Decision Letter 0]

24 Mar 2026

PNTD-D-26-00281

Performance of Fecal Inflammatory Biomarkers to Identify Watery Shigellosis: Findings from the Enterics for Global Health (EFGH) Shigella Surveillance StudyPLOS Neglected Tropical DiseasesDear Dr. Ogwel,Thank you for submitting your manuscript to PLOS Neglected Tropical Diseases. After careful consideration, we feel that it has merit but does not fully meet PLOS Neglected Tropical Diseases's publication criteria as it currently stands. Therefore, we invite you to submit a revised version of the manuscript that addresses the points raised during the review process.Please submit your revised manuscript by May 23 2026 11:59PM. If you will need more time than this to complete your revisions, please reply to this message or contact the journal office at plosntds@plos.org. Please include the following items when submitting your revised manuscript:* A letter that responds to each point raised by the editor and reviewer(s). You should upload this letter as a separate file labeled 'Response to Reviewers'. This file does not need to include responses to any formatting updates and technical items listed in the 'Journal Requirements' section below.* A marked-up copy of your manuscript that highlights changes made to the original version. You should upload this as a separate file labeled 'Revised Manuscript with Track Changes'.* An unmarked version of your revised paper without tracked changes. You should upload this as a separate file labeled 'Manuscript'.If you would like to make changes to your financial disclosure, competing interests statement, or data availability statement, please make these updates within the submission form at the time of resubmission. Guidelines for resubmitting your figure files are available below the reviewer comments at the end of this letter.We look forward to receiving your revised manuscript.Kind regards,Abiola Senok, MBBS; PhD, FRCPathAcademic EditorPLOS Neglected Tropical DiseasesStuart BlacksellSection EditorPLOS Neglected Tropical Diseases

Shaden Kamhawi

co-Editor-in-Chief

Paul Brindley

co-Editor-in-Chief

**Journal Requirements:**

At this stage, the following Authors/Authors require contributions: Sara Kim. Please ensure that the full contributions of each author are acknowledged in the "Add/Edit/Remove Authors" section of our submission form.

2) Please amend your detailed Financial Disclosure statement. This is published with the article. It must therefore be completed in full sentences and contain the exact wording you wish to be published.

State the initials, alongside each funding source, of each author to receive each grant. For example: "This work was supported by the National Institutes of Health (####### to AM; ###### to CJ) and the National Science Foundation (###### to AM).".

**Reviewers' comments:**Reviewer's Responses to Questions

**Key Review Criteria Required for Acceptance?**

**Methods**

-Are the objectives of the study clearly articulated with a clear testable hypothesis stated?

-Is the study design appropriate to address the stated objectives?

-Is the population clearly described and appropriate for the hypothesis being tested?

-Is the sample size sufficient to ensure adequate power to address the hypothesis being tested?

-Were correct statistical analysis used to support conclusions?

-Are there concerns about ethical or regulatory requirements being met?

Reviewer #1: Bacterial diarrhea, particularly that caused by Shigella, is typically an invasive infection characterized by the presence of blood in the stool. However, in some cases, patients with invasive bacterial diarrhea may not have visibly bloody stools. This study aims to identify differences among patients with diarrhea (without visible blood in the stool) in order to better inform clinical practice. I believe this point should be emphasized in the background. Alternatively, in line 55, the phrase “watery diarrhea” could be revised to “watery diarrhea (non-bloody)” to clarify the study objective more clearly.

In the predictors section:

- The authors could define dehydration status at line 235 (i.e., severe dehydration and some dehydration, mentioned at line 325).

- The authors use the variable “Site-specific indicator of the Shigella transmission season (defined as the 3 months with the highest Shigella prevalence at each site)” as a predictor. The authors may explain the rationale for selecting this variable. Is diarrhea caused by Shigella a disease with clear seasonal variation? Additionally, is there any difference in the timing of Shigella diarrhea cases? For example, are there differences in the months with the highest number of Shigella diarrhea cases among the six cities included in the study? Furthermore, does the peak period of Shigella diarrhea vary from year to year? If so, it may be difficult to use this variable as a reliable predictor of the causative pathogen.

Reviewer #2: No new analyses are required for acceptance. The objectives of the study are clearly stated; the study design, sample size, and statistics are appropriate; and there are no ethical or regulatory concerns.

**Results**

-Does the analysis presented match the analysis plan?

-Are the results clearly and completely presented?

-Are the figures (Tables, Images) of sufficient quality for clarity?

Reviewer #1: In this study, 55.8% of participants were male; therefore, I think the use of the word “most” in line 323 may not be appropriate. It may be more accurate to state that “more than half were male.”

In Table 1:

- The authors may consider adding p-values to compare the characteristics across the six cities.

- Regarding age groups, the authors may also explain the rationale for categorizing age as 6–11, 12–17, 18–23, and 24–35 months, rather than 6–11, 12–17, and 18–35 months as presented in Table 4.

Reviewer #2: The results presented match the analysis plan and the results are clearly presented. However, the figures had low resolution, but that could be due to the PDF generated from the files. I also did not see any figure legends included in the manuscript text or attached with the figures (for figures 1-3 in the manuscript).

**Conclusions**

-Are the conclusions supported by the data presented?

-Are the limitations of analysis clearly described?

-Do the authors discuss how these data can be helpful to advance our understanding of the topic under study?

-Is public health relevance addressed?

Reviewer #1: (No Response)

Reviewer #2: The conclusions are supported by the data presented, and the limitations of the study are clearly described. The authors discuss how the study contributes to the topic and is relevant to public health.

**Editorial and Data Presentation Modifications?**

Reviewer #1: (No Response)

Reviewer #2: (No Response)

**Summary and General Comments**

Reviewer #1: (No Response)

Reviewer #2: The manuscript highlights analysis of fecal inflammatory biomarkers to identify watery shigellosis. Fecal samples were collected across six established study sites and analyzed in children aged 6-35 months presenting with watery diarrhea. Fecal samples were analyzed by culturing or qPCR, and for the presence of several biomarkers of inflammatory diarrhea. A model was then generated to predict watery diarrhea as a result of Shigella infection. Several iterations of the model were tested, but the full model incorporating all predictors of watery diarrhea had the best results. The authors conclude that one biomarker in particular, fecal hemoglobin, improved the prediction scores. Consideration of fecal hemoglobin could improve diagnostics and ultimately treatment for shigellosis.

Overall, the authors present a thorough study with important reference standards for the data; describe the methodology, outcomes, and model generation well; and address limitations. Overall, the model performs well, especially when considering that the authors aim to detect Shigella based on watery diarrhea and not when dysentery (inflammatory diarrhea) is present. Here is where a significant strength of the study is noted, since the impact of Shigella infections are often limited by variability in symptoms and difficult in detecting the pathogen by culture or qPCR. Thus, this model may help with clinical diagnosis and ultimately treatment decisions. The manuscript is presented well, and no major weaknesses were identified. However, a few minor text edits are suggested for clarifications or to further enhance the manuscript, especially more general readers:

1. Is antibiotic treatment restricted with watery diarrhea since it can resolve with supportive care? Or is the issue of limitation due to unknown etiology and/or concerns about AMR?

2. For studies like this one presented, what is considered a good or excellent AUC? The authors discuss other studies and compare their results well, but a statement of a good to excellent AUC range would be helpful.

3. Could the calculated AUC have been lowered due to false negatives with rectal swab culturing or qPCR?

4. Since Shigella diagnosis can be difficult for multiple reasons, this approach also helpful for surveillance and more thoroughly understanding the global impact of Shigella.

PLOS authors have the option to publish the peer review history of their article (what does this mean?). If published, this will include your full peer review and any attached files.

Reviewer #1: No

Reviewer #2: No

**Figure resubmission:**While revising your submission, we strongly recommend that you use PLOS’s NAAS tool (https://ngplosjournals.pagemajik.ai/artanalysis) to test your figure files. NAAS can convert your figure files to the TIFF file type and meet basic requirements (such as print size, resolution), or provide you with a report on issues that do not meet our requirements and that NAAS cannot fix.

After uploading your figures to PLOS’s NAAS tool - https://ngplosjournals.pagemajik.ai/artanalysis, NAAS will process the files provided and display the results in the "Uploaded Files" section of the page as the processing is complete. If the uploaded figures meet our requirements (or NAAS is able to fix the files to meet our requirements), the figure will be marked as "fixed" above. If NAAS is unable to fix the files, a red "failed" label will appear above. When NAAS has confirmed that the figure files meet our requirements, please download the file via the download option, and include these NAAS processed figure files when submitting your revised manuscript.**Reproducibility:**To enhance the reproducibility of your results, we recommend that authors of applicable studies deposit laboratory protocols in protocols.io, where a protocol can be assigned its own identifier (DOI) such that it can be cited independently in the future. Additionally, PLOS ONE offers an option to publish peer-reviewed clinical study protocols. Read more information on sharing protocols at https://plos.org/protocols?utm_medium=editorial-email&utm_source=authorletters&utm_campaign=protocols

---

## [Decision Letter · Decision Letter 1]

6 May 2026

Dear Dr Ogwel,

We are pleased to inform you that your manuscript 'Performance of Fecal Inflammatory Biomarkers to Identify Watery Shigellosis: Findings from the Enterics for Global Health (EFGH) Shigella Surveillance Study' has been provisionally accepted for publication in PLOS Neglected Tropical Diseases.

Best regards,

Abiola Senok, MBBS; PhD, FRCPath

Academic Editor

Stuart Blacksell

Section Editor

Shaden Kamhawi

co-Editor-in-Chief

Paul Brindley

co-Editor-in-Chief

Reviewer's Responses to Questions

**Key Review Criteria Required for Acceptance?**

**Methods**

-Are the objectives of the study clearly articulated with a clear testable hypothesis stated?

-Is the study design appropriate to address the stated objectives?

-Is the population clearly described and appropriate for the hypothesis being tested?

-Is the sample size sufficient to ensure adequate power to address the hypothesis being tested?

-Were correct statistical analysis used to support conclusions?

-Are there concerns about ethical or regulatory requirements being met?

Reviewer #1: (No Response)

Reviewer #2: (No Response)

**Results**

-Does the analysis presented match the analysis plan?

-Are the results clearly and completely presented?

-Are the figures (Tables, Images) of sufficient quality for clarity?

Reviewer #1: (No Response)

Reviewer #2: (No Response)

**Conclusions**

-Are the conclusions supported by the data presented?

-Are the limitations of analysis clearly described?

-Do the authors discuss how these data can be helpful to advance our understanding of the topic under study?

-Is public health relevance addressed?

Reviewer #1: (No Response)

Reviewer #2: (No Response)

**Editorial and Data Presentation Modifications?**

Reviewer #1: (No Response)

Reviewer #2: (No Response)

**Summary and General Comments**

Reviewer #1: (No Response)

Reviewer #2: The authors have addressed the critiques from the previous review and have revised the manuscript accordingly. I have no additional concerns and maintain that the authors presented an important and sound study with appropriate analyses and conclusions based on the data.

PLOS authors have the option to publish the peer review history of their article (what does this mean?). If published, this will include your full peer review and any attached files.

Reviewer #1: No

Reviewer #2: No

---

## [Editor Report · Acceptance letter]

Dear Dr Ogwel,

We are delighted to inform you that your manuscript, "Performance of Fecal Inflammatory Biomarkers to Identify Watery Shigellosis: Findings from the Enterics for Global Health (EFGH) Shigella Surveillance Study," has been formally accepted for publication in PLOS Neglected Tropical Diseases.

Best regards,

Shaden Kamhawi

co-Editor-in-Chief

Paul Brindley

co-Editor-in-Chief
